# Why Diffusion Models Don't Memorize: The Role of Implicit Dynamical Regularization in Training

**Tony Bonnaire**[†]
LPENS
Université PSL, Paris
`tony.bonnaire@phys.ens.fr`

**Raphaël Urfin**[†]
LPENS
Université PSL, Paris
`raphael.urfin@phys.ens.fr`

**Giulio Biroli**
LPENS
Université PSL, Paris
`giulio.biroli@phys.ens.fr`

**Marc Mézard**
Department of Computing Sciences
Bocconi University, Milano
`marc.mezard@unibocconi.it`

## Abstract

Diffusion models have achieved remarkable success across a wide range of generative tasks. A key challenge is understanding the mechanisms that prevent their memorization of training data and allow generalization. In this work, we investigate the role of the training dynamics in the transition from generalization to memorization. Through extensive experiments and theoretical analysis, we identify two distinct timescales: an early time $\tau_{\mathrm{gen}}$ at which models begin to generate high-quality samples, and a later time $\tau_{\mathrm{mem}}$ beyond which memorization emerges. Crucially, we find that $\tau_{\mathrm{mem}}$ increases linearly with the training set size $n$, while $\tau_{\mathrm{gen}}$ remains constant. This creates a growing window of training times with $n$ where models generalize effectively, despite showing strong memorization if training continues beyond it. It is only when $n$ becomes larger than a model-dependent threshold that overfitting disappears at infinite training times. These findings reveal a form of implicit dynamical regularization in the training dynamics, which allow to avoid memorization even in highly overparameterized settings. Our results are supported by numerical experiments with standard U-Net architectures on realistic and synthetic datasets, and by a theoretical analysis using a tractable random features model studied in the high-dimensional limit.

## 1 Introduction

Diffusion Models [DMs, 44, 18, 49, 50] achieve state-of-the-art performance in a wide variety of AI tasks such as the generation of images [41], audios [58], videos [30], and scientific data [28, 36]. This class of generative models, inspired by out-of-equilibrium thermodynamics [44], corresponds to a two-stage process: the first one, called *forward*, gradually adds noise to a data, whereas the second one, called *backward*, generates new data by denoising Gaussian white noise samples. In DMs, the reverse process typically involves solving a stochastic differential equation (SDE) with a force field called *score*. However, it is also possible to define a deterministic transport through an ordinary differential equation (ODE), treating the score as a velocity field, an approach that is for instance followed in flow matching [29].

Understanding the generalization properties of score-based generative methods is a central issue in machine learning, and a particularly important question is how memorization of the training set

---

[†]Equal contribution.

39th Conference on Neural Information Processing Systems (NeurIPS 2025).

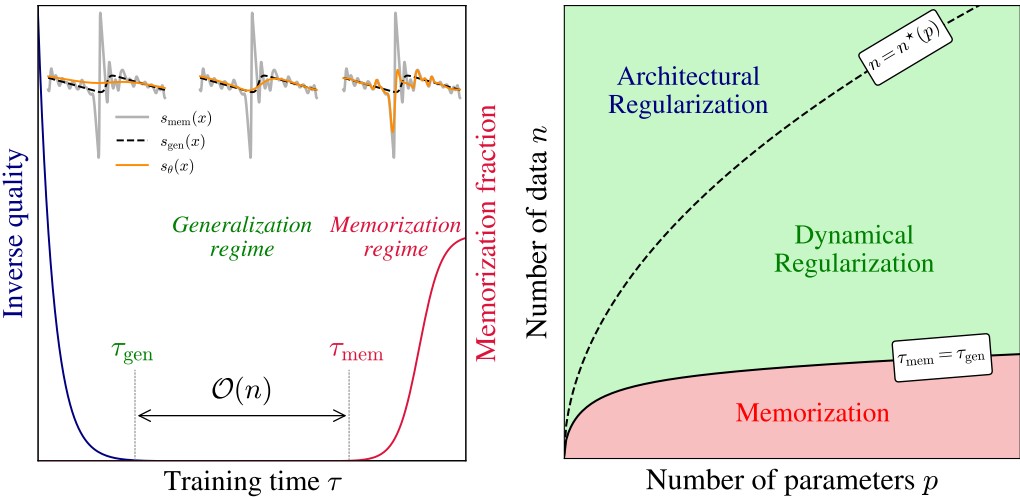

Figure 1: **Qualitative summary of our contributions.** *(Left)* Illustration of the training dynamics of a diffusion model. Depending on the training time $\tau$, we identify three regimes measured by the inverse quality of the generated samples (blue curve) and their memorization fraction (red curve). The generalization regime extends over a large window of training times which increases with the training set size $n$. On top, we show a one dimensional example of the learned score function during training (orange). The gray line gives the exact empirical score, at a given noise level, while the black dashed line corresponds to the true (population) score. *(Right)* Phase diagram in the $(n, p)$ plane illustrating three regimes of diffusion models: Memorization when $n$ is sufficiently small at fixed $p$, Architectural Regularization for $n > n^\star(p)$ (which is model and dataset dependent, as discussed in [13, 23]), and Dynamical Regularization, corresponding to a large intermediate generalization regime obtained when the training dynamics is stopped early, i.e. $\tau \in [\tau_{\text{gen}}, \tau_{\text{mem}}]$.

is avoided in practice. A model without regularization achieving zero training loss only learns the empirical score, and is bound to reproduce samples of the training dataset at the end of the backward process. This memorization regime [27, 5, 4] is empirically observed when the training set is small and disappears when it increases beyond a model-dependent threshold [22]. Understanding the mechanisms controlling this change of regimes from memorization to generalization is a central challenge for both theory and applications. Model regularization and inductive biases imposed by the network architecture were shown to play a role [23, 43], as well as a dynamical regularization due to the finiteness of the learning rate [56]. However, the regime shift described above is consistently observed even in models where all these regularization mechanisms are present. This suggests that the core mechanism behind the transition from memorization to generalization lies elsewhere. In this work, we demonstrate – first through numerical experiments, and then via the theoretical analysis of a simplified model – that this transition is driven by an implicit dynamical bias towards generalizing solutions emerging in the training, which allows to avoid the memorization phase.

**Contributions and theoretical picture.** We investigate the dynamics of score learning using gradient descent, both numerically and analytically, and study the generation properties of the score depending on the time $\tau$ at which the training is stopped. The theoretical picture built from our results and combining several findings from the recent literature is illustrated in Fig. 1. The two main parameters are the size of the training set $n$ and the expressivity of the class of score functions on which one trains the model, characterized by a number of parameters $p$; when both $n$ and $p$ are large one can identify three main regimes. Given $p$, if $n$ is larger than $n^*(p)$ (which depends on the training set and on the class of scores), the score model is not expressive enough to represent the empirical score associated to $n$ data, and instead provides a smooth interpolation, approximately independent of the training set. In this regime, even with a very large training time $\tau \to \infty$, memorization does not occur because the model is regularized by its architecture and the finite number of parameters. When $n < n^*(p)$ the model is expressive enough to memorize, and two timescales emerge during training: one, $\tau_{\text{gen}}$, is the minimum training time required to achieve high-quality data generation; the second, $\tau_{\text{mem}} > \tau_{\text{gen}}$, signals when further training induces memorization, and causes the model to

increasingly reproduce the training samples (left panel). The first timescale, $\tau_{\mathrm{gen}}$, is found independent of $n$, whereas the second, $\tau_{\mathrm{mem}}$, grows approximately linearly with $n$, thus opening a large window of training times during which the model generalizes if early stopped when $\tau \in [\tau_{\mathrm{gen}}, \tau_{\mathrm{mem}}]$. Our results shows that implicit dynamical regularization in training plays a crucial role in score-based generative models, substantially enlarging the generalization regime (see right panel of Fig. 1), and hence allowing to avoid memorization even in highly overparameterized settings. We find that the key mechanism behind the widening gap between $\tau_{\mathrm{gen}}$ and $\tau_{\mathrm{mem}}$ is the irregularity of the empirical score at low noise level and large $n$. In this regime the models used to approximate the score provide a smooth interpolation that remains stable for a long period of training times and closely approximates the population score, a behavior likely rooted in the spectral bias of neural networks [38]. Only at very long training times do the dynamics converge to the low lying minimum corresponding to the empirical score, leading to memorization (as illustrated in the 1D examples in the left panel of Fig. 1).

The theoretical picture described above is based on our numerical and analytical results, and builds up on previous works, in particular numerical analysis characterizing the memorization–generalization transition [16, 57], analytical works on memorization of DMs [13, 23, 22], and studies on the spectral bias of deep neural networks [38]. Our numerical experiments[†] use a class of scores based on a realistic U-Net [42] trained on downscaled images of the CelebA dataset [31]. By varying $n$ and $p$, we measure the evolution of the sample quality (through FID) and the fraction of memorization during learning, which support the theoretical scenario presented in Fig. 1. Additional experimental results on synthetic data are provided in Supplemental Material (SM, Sects. A and B). On the analytical side, we focus on a class of scores constructed from random features and simplified models of data, following [13]. In this setting, the timescales of training dynamics correspond directly to the inverse eigenvalues of the random feature correlation matrix. Leveraging tools from random matrix theory, we compute the spectrum in the limit of large datasets, high-dimensional data, and overparameterized models. This analysis reveals, in a fully tractable way, how the theoretical picture of Fig. 1 emerges within the random feature framework.

**Related works.**

- The memorization transition in DMs has been the subject of several recent empirical investigations [8, 45, 46] which have demonstrated that state-of-the-art image DMs – including Stable Diffusion and DALL·E – can reproduce a non-negligible portion of their training data, indicating a form of memorization. Several additional works [16, 57] examined how this phenomenon is influenced by factors such as data distribution, model configuration, and training procedure, and provide a strong basis for the numerical part of our work.

- A series of theoretical studies in the high-dimensional regime have analyzed the memorization–generalization transition during the generative dynamics under the empirical score assumption [5, 1, 52], showing how trajectories are attracted to the training samples. Within this high-dimensional framework, [9, 10, 55, 13] study the score learning for various model classes. In particular, [13] uses a Random Feature Neural Network [39]. The authors compute the asymptotic training and test losses for $\tau \to \infty$ and relate it to memorization. The theoretical part of our work generalizes this approach to study the role of training dynamics and early stopping in the memorization–generalization transition.

- Recent works have also uncovered complementary sources of implicit regularization explaining how DMs avoid memorization. Architectural biases and limited network capacity were for instance shown to constrain memorization in [23, 22, 4], and finiteness of the learning rate prevents the model from learning the empirical score in [56]. Also related to our analysis, [26, 4] show the beneficial role of early stopping the training dynamics to enhance the generalization.

- Finally, previous studies on supervised learning [38, 59], and more recently on DMs [54], have shown that deep neural networks display a frequency-dependent learning speed, and hence a learning bias towards low frequency functions. This fact plays an important role in the results we present since the empirical score contains a low frequency part that is close to the population score, and a high-frequency part that is dataset-dependent. To the best of our knowledge, the training time to learn the high-frequency part and hence memorize, that we find to scale with $n$, has not been studied from this perspective in the context of score-based generative methods.

---

[†]Code available at github.com/tbonnair/Why-Diffusion-Models-Don-t-Memorize.

**Setting: generative diffusion and score learning.** Standard DMs define a transport from a target distribution $P_0$ in $\mathbb{R}^d$ to a Gaussian white noise $\mathcal{N}(0, \boldsymbol{I}_d)$ through a *forward process* defined as an Ornstein-Uhlenbeck (OU) stochastic differential equation (SDE):

$$d\mathbf{x} = -\mathbf{x}(t)dt + d\mathbf{B}(t), \tag{1}$$

where $d\mathbf{B}(t)$ is square root of two times a Wiener process. Generation is performed by time-reversing the SDE (1) using the score function $\mathbf{s}(\mathbf{x}, t) = \nabla_{\mathbf{x}} \log P_t(\mathbf{x})$,

$$-d\mathbf{x} = [\mathbf{x}(t) + 2\mathbf{s}(\mathbf{x}, t)] \, dt + d\mathbf{B}(t), \tag{2}$$

where $P_t(\mathbf{x})$ is the probability density at time $t$ along the forward process, and the noise $d\mathbf{B}(t)$ is also the square root of two times a Wiener process. As shown in the seminal works [21, 53], $\mathbf{s}(\mathbf{x}, t)$ can be obtained by minimizing the score matching loss

$$\hat{\mathbf{s}}(\mathbf{x}, t) = \arg \min_{\mathbf{s}} \mathbb{E}_{\mathbf{x} \sim P_0, \boldsymbol{\xi} \sim \mathcal{N}(0, \boldsymbol{I}_d)} \left[ \| \sqrt{\Delta_t} \mathbf{s}(\mathbf{x}(t), t) + \boldsymbol{\xi} \|^2 \right], \tag{3}$$

where $\Delta_t = 1 - e^{-2t}$. In practice, the optimization problem is restricted to a parametrized class of functions $\mathbf{s}_{\boldsymbol{\theta}}(\mathbf{x}(t), t)$ defined, for example, by a neural network with parameters $\boldsymbol{\theta}$. The expectation over $\mathbf{x}$ is replaced by the empirical average over the training set ($n$ iid samples $\mathbf{x}^{\nu}$ drawn from $P_0$),

$$\mathcal{L}_t(\boldsymbol{\theta}, \{\mathbf{x}^{\nu}\}_{\nu=1}^n) = \frac{1}{n} \sum_{\nu=1}^{n} \mathbb{E}_{\boldsymbol{\xi} \sim \mathcal{N}(0, \boldsymbol{I}_d)} \left[ \| \sqrt{\Delta_t} \mathbf{s}_{\boldsymbol{\theta}}(\mathbf{x}^{\nu}(t)) + \boldsymbol{\xi} \|^2 \right], \tag{4}$$

where $\mathbf{x}_t^{\nu}(\boldsymbol{\xi}) = e^{-t} \mathbf{x}^{\nu} + \sqrt{\Delta_t} \boldsymbol{\xi}$. The loss in (4) can be minimized with standard optimizers, such as stochastic gradient descent [SGD, 40] or Adam [25]. In practice, a single model conditioned on the diffusion time $t$ is trained by integrating (4) over time [24]. The solution of the minimization of (4) is the so-called empirical score (e.g. [5, 27]), defined as $\mathbf{s}_{\text{emp}}(\mathbf{x}, t) = \nabla_{\mathbf{x}} \log P_t^{\text{emp}}(\mathbf{x})$, with

$$P_t^{\text{emp}}(\mathbf{x}) = \frac{1}{n \left(2\pi \Delta_t\right)^{d/2}} \sum_{\nu=1}^{n} e^{-\frac{1}{2\Delta_t} \|\mathbf{x} - \mathbf{x}^{\nu} e^{-t}\|_2^2}. \tag{5}$$

This solution is known to inevitably recreate samples of the training set at the end of the generative process (i.e., it perfectly memorizes), unless $n$ grows exponentially with the dimension $d$ [5]. However, this is not the case in many practical applications where memorization is only observed for relatively small values of $n$, and disappears well before $n$ becomes exponentially large in $d$. The empirical minimization performed in practice, within a given class of models and a given minimization procedure, does *not* drive the optimization to the global minimum of (4), but instead to a smoother estimate of the score that is independent of the training set with good generalization properties [22], as the global minimum of (3) would do. Understanding how it is possible, and in particular the role played by the training dynamics to avoid memorization, is the central aim of the present work.

## 2 Generalization and memorization during training of diffusion models

**Data & architecture.** We conduct our experiments on the CelebA face dataset [31], which we convert to grayscale downsampled images of size $d = 32 \times 32$, and vary the training set size $n$ from 128 up to 32768. Our score model has a U-Net architecture [42] with three resolution levels and a base channel width of $W$ with multipliers 1, 2 and 3 respectively. All our networks are DDPMs [18] trained to predict the injected noise at diffusion time $t$ using SGD with momentum at fixed batch size $\min(n, 512)$. The models are all conditioned on $t$, i.e. a single model approximates the score at all times, and make use of a standard sinusoidal position embedding [51] that is added to the features of each resolution. More details about the numerical setup can be found in SM (Sect. A).

**Evaluation metrics.** To study the transition from generalization to memorization during training, we monitor the loss (4) during training using a fixed diffusion time $t = 0.01$. At various numbers of SGD updates $\tau$, we compute the loss on all $n$ training examples (training loss) and on a held-out test set of 2048 images (test loss). To characterize the score obtained after a training time $\tau$, we assess the originality and quality of samples by generating 10K samples using a DDIM accelerated sampling [47]. We compute (i) the Fréchet-Inception Distance [FID, 17] against 10K test samples which we use to identify the generalization time $\tau_{\text{gen}}$; and (ii) the fraction of memorized generated

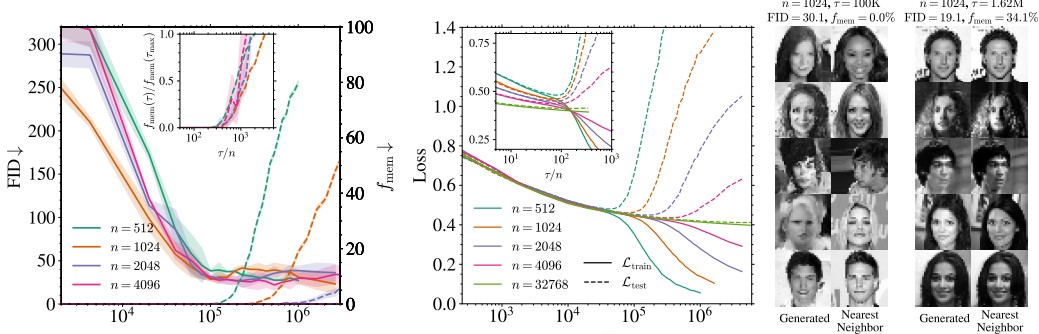

Figure 2: **Memorization transition as a function of the training set size $n$ for U-Net score models on CelebA.** *(Left)* FID (solid lines, left axis) and memorization fraction $f_{\mathrm{mem}}$ (dashed lines, right axis) against training time $\tau$ for various $n$. Inset: normalized memorization fraction $f_{\mathrm{mem}}(\tau)/f_{\mathrm{mem}}(\tau_{\max})$ with the rescaled time $\tau/n$. *(Middle)* Training (solid lines) and test (dashed lines) loss with $\tau$ for several $n$ at fixed $t = 0.01$. Inset: both losses plotted against $\tau/n$. Error bars on the losses are imperceptible. *(Right)* Generated samples from the model trained with $n = 1024$ for $\tau = 100\mathrm{K}$ or $\tau = 1.62\mathrm{M}$ steps, along with their nearest neighbors in the training set.

samples $f_{\mathrm{mem}}(\tau)$ granting access to $\tau_{\mathrm{mem}}$, the memorization time. Following previous numerical studies [57, 16], a generated sample $\mathbf{x}_\tau$ is considered memorized if

$$\mathbb{E}_{\mathbf{x}_\tau}\left[\frac{\|\mathbf{x}_\tau - \mathbf{a}^{\mu_1}\|_2}{\|\mathbf{x}_\tau - \mathbf{a}^{\mu_2}\|_2}\right] < k, \tag{6}$$

where $\mathbf{a}^{\mu_1}$ and $\mathbf{a}^{\mu_2}$ are the nearest and second nearest neighbors of $\mathbf{x}_\tau$ in the training set in the $L_2$ sense. In what follows, we choose to work with $k = 1/3$ [57, 16], but we checked that varying $k$ to $1/2$ or $1/4$ does not impact the claims about the scaling. Error bars in the figures correspond to twice the standard deviation over 5 different test sets for FIDs, and 5 noise realizations for $\mathcal{L}_{\mathrm{train}}$ and $\mathcal{L}_{\mathrm{test}}$. For $f_{\mathrm{mem}}$, we report the 95% CIs on the mean evaluated with 1,000 bootstrap samples.

**Role of training set size on the learning dynamics.** At fixed model capacity ($p = 4 \times 10^6$, base width $W = 32$), we investigate how the training set size $n$ impacts the previous metrics. In the left panel of Fig. 2, we first report the FID (solid lines) and $f_{\mathrm{mem}}(\tau)$ (dashed lines) for various $n$. All trainings dynamics exhibit two phases. First, the FID quickly decreases to reach a minimum value on a timescale $\tau_{\mathrm{gen}}$ ($\approx 100\mathrm{K}$) that does not depend on $n$. In the right panel, the generated samples at $\tau = 100\mathrm{K}$ clearly differ from their nearest neighbors in the training set, indicating that the model generalizes correctly. Beyond this time, the FID remains flat. $f_{\mathrm{mem}}(\tau)$ is zero until a later time $\tau_{\mathrm{mem}}$ after which it increases, clearly signaling the entrance into a memorization regime, as illustrated by the generated samples in the right-most panel of Fig. 2, very close to their nearest neighbors. Both the transition time $\tau_{\mathrm{mem}}$ and the value of the final fraction $f_{\mathrm{mem}}(\tau_{\max})$ (with $\tau_{\max}$ being one to four million SGD steps) vary with $n$. The inset plot shows the normalized memorization fraction $f_{\mathrm{mem}}(\tau)/f_{\mathrm{mem}}(\tau_{\max})$ against the rescaled time $\tau/n$, making all curves collapse and increase at around $\tau/n \approx 300$, showing that $\tau_{\mathrm{mem}} \propto n$, and demonstrating the existence of a generalization window for $\tau \in [\tau_{\mathrm{gen}}, \tau_{\mathrm{mem}}]$ that widens linearly with $n$, as illustrated in the left panel of Fig. 1.

As highlighted in the introduction, memorization in DMs is ultimately driven by the overfitting of the empirical score $\mathbf{s}_{\mathrm{mem}}(\mathbf{x}, t)$. The evolution of $\mathcal{L}_{\mathrm{train}}(\tau)$ and $\mathcal{L}_{\mathrm{test}}(\tau)$ at fixed $t = 0.01$ are shown in the middle panel of Fig. 2 for $n$ ranging from 512 to 32768. Initially, the two losses remain nearly indistinguishable, indicating that the learned score $\mathbf{s}_{\boldsymbol{\theta}}(\mathbf{x}, t)$ does not depend on the training set. Beyond a critical time, $\mathcal{L}_{\mathrm{train}}$ continues to decrease while $\mathcal{L}_{\mathrm{test}}$ increases, leading to a nonzero generalization loss whose magnitude depends on $n$. As $n$ increases, this critical time also increases and, eventually, the training and test loss gap shrinks: for $n = 32768$, the test loss remains close to the training loss, even after 11 million SGD steps. The inset shows the evolution of both losses with $\tau/n$, demonstrating that the overfitting time scales linearly with the training set size $n$, just like $\tau_{\mathrm{mem}}$ identified in the left panel. Moreover, there is a consistent lag between the overfitting time and $\tau_{\mathrm{mem}}$ at fixed $n$, reflecting the additional training required for the model to overfit the empirical score sufficiently to reproduce the training samples, and therefore to impact the memorization fraction.

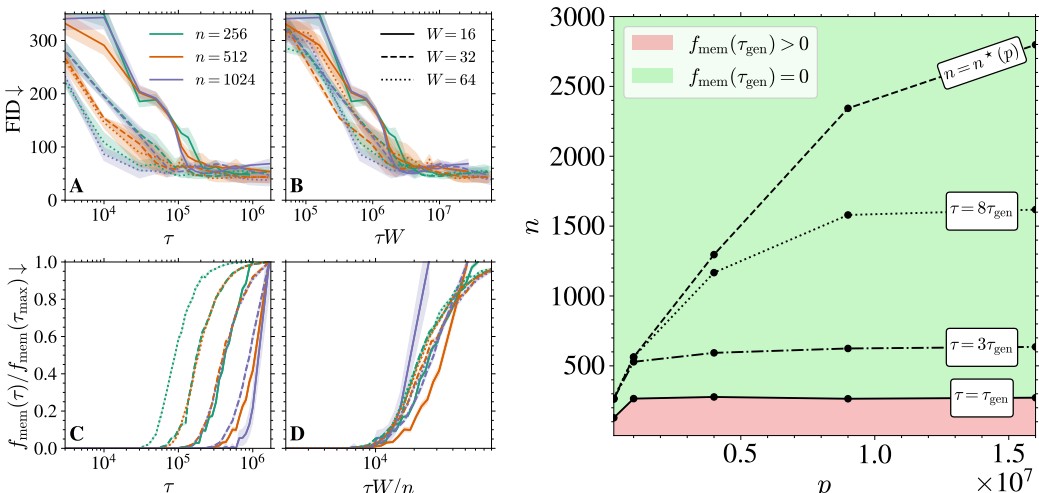

Figure 3: **Effect of the number of parameters in the U-Net architecture on the timescales of the training dynamics.** *(Left)* FID (panels **A**, **B**) and normalized memorization fraction $f_{\mathrm{mem}}(\tau)/f_{\mathrm{mem}}(\tau_{\max})$ (panels **C**, **D**) for various $n$ and $W$ during training. In panels **B** and **D**, time is rescaled such that all curves collapse. *(Right)* $(n, p)$ phase diagram of generalization vs memorization for U-Nets trained on CelebA. Curves show, for $\tau \in \{\tau_{\mathrm{gen}}, 3\tau_{\mathrm{gen}}, 8\tau_{\mathrm{gen}}\}$, the minimal dataset size $n(p)$ satisfying $f_{\mathrm{mem}}(\tau) = 0$. The shaded background indicates the memorization–generalization boundary for $\tau = \tau_{\mathrm{gen}}$.

**Memorization is *not* due to data repetition.**    We must stress that this delayed memorization with $n$ is *not* due to the mere repetition of training samples, as a first intuition could suggest. In SM Sects. A and B, we show that full-batch updates still yield $\tau_{\mathrm{mem}} \propto n$. In other words, even if at fixed $\tau$ all models have processed each sample equally often, larger $n$ consistently postpone memorization. This confirms that memorization in DMs is driven by a fundamental $n$-dependent change in the loss landscape – not by a sample repetition during training.

**Effect of the model capacity.**    To study more precisely the role of the model capacity on the memorization–generalization transition, we vary the number of parameters $p$ by changing the U-Nets base width $W \in \{8, 16, 32, 48, 64\}$, resulting in a total of $p \in \{0.26, 1, 4, 9, 16\} \times 10^6$ parameters. In the left panel of Fig. 3, we plot both the FID (top row) and the normalized memorization fraction (bottom row) as functions of $\tau$ for several width $W$ and training set sizes $n$. Panels **A** and **C** demonstrate that higher-capacity networks (larger $W$) achieve high-quality generation and begin to memorize *earlier* than smaller ones. Panels **B** and **D** show that the two characteristic timescales simply scale as $\tau_{\mathrm{gen}} \propto W^{-1}$ and $\tau_{\mathrm{mem}} \propto nW^{-1}$. In particular, this implies that, for $W > 8$, the critical training set size $n_{\mathrm{gm}}(p)$ at which $\tau_{\mathrm{mem}} = \tau_{\mathrm{gen}}$ is approximately independent of $p$ (at least on the limited values of $p$ we focused on).When $n > n_{\mathrm{gm}}(p)$, the interval $[\tau_{\mathrm{gen}}, \tau_{\mathrm{mem}}]$ opens up, so that early stopping within this window yields high quality samples without memorization. In the right panel of Fig. 3, we display this boundary (solid line) in the $(n, p)$ plane by fixing the training time to $\tau = \tau_{\mathrm{gen}}$, that we identify numerically using the collapse of all FIDs at around $W\tau_{\mathrm{gen}} \approx 3 \times 10^6$ (see panel **B**), and computing the smallest $n$ such that $f_{\mathrm{mem}}(\tau) = 0$. The resulting solid curve delineates two regimes: below the curve, memorization already starts at $\tau_{\mathrm{gen}}$; above the curve, the models generalize perfectly under early stopping. We repeat this experiment for $\tau = 3\tau_{\mathrm{gen}}$ and $\tau = 8\tau_{\mathrm{gen}}$, showing saturation to larger and larger $p$ as $\tau$ increases. Eventually, for $\tau \to \infty$, we expect these successive boundaries to converge to the architectural regularization threshold $n^\star(p)$, i.e. the point beyond which the network avoids memorization because it is not expressive enough, as found in [13] and highlighted in the right panel of Fig. 1. In order to estimate $n^\star(p)$, we measure for a given $\tau$ the largest $n(\tau)$ yielding $f_{\mathrm{mem}} \approx 0$. The curve $n(\tau)$ approaches $n^\star(p)$ for large $\tau$. We therefore estimate $n^\star(p)$ by measuring the asymptotic values of $n(\tau)$, which in practice is reached already at $\tau = \tau_{\max} = 2M$ updates for the values of $W$ we focus on.

## 3 Training dynamics of a Random Features Network

**Notations.** We use bold symbols for vectors and matrices. The $L^2$ norm of a vector $\mathbf{x}$ is denoted by $\|\mathbf{x}\| = (\sum_i \mathbf{x}_i^2)^{1/2}$. We write $f = \mathcal{O}(g)$ to mean that in the limit $n, p \to \infty$, there exists a constant $C$ such that $|f| \leq C|g|$.

**Setting.** We study analytically a model introduced in [13], where the data lie in $d$ dimensions. We parametrize the score with a Random Features Neural Network [RFNN, 39]

$$\mathbf{s_A}(\mathbf{x}) = \frac{\mathbf{A}}{\sqrt{p}} \sigma\left(\frac{\mathbf{Wx}}{\sqrt{d}}\right). \tag{7}$$

An RFNN, illustrated in Fig. 4 (left), is a two-layer neural-network whose first layer weights ($\mathbf{W} \in \mathbb{R}^{p \times d}$) are drawn from a Gaussian distribution and remain frozen while the second layer weights ($\mathbf{A} \in \mathbb{R}^{d \times p}$) are learned during training. This model has already served as theoretical framework for studying several behaviors of deep neural network such as the double descent phenomenon [32, 11]. $\sigma$ is an element-wise non-linear activation function. We consider a training set of $n$ iid samples $\mathbf{x}^\nu \sim P_\mathbf{x}$ for $\nu = 1, \dots, n$ and we focus on the high-dimensional limit $d, p, n \to \infty$ with the ratios $\psi_p = p/d, \psi_n = n/d$ kept fixed. We study the training dynamics associated to the minimization of the empirical score matching loss defined in (4) at a fixed diffusion time $t$. This is a simplification compared to practical methods, which use a single model for all $t$. It has been already studied in previous theoretical works [9, 13]. The loss (4) is rescaled by a factor $1/d$ in order to ensure a finite limit at large $d$. We also study the evolution of the test loss evaluated on test points and the distance to the exact score $\mathbf{s}(\mathbf{x}) = \nabla \log P_\mathbf{x}$,

$$\mathcal{L}_{\text{test}} = \frac{1}{d}\mathbb{E}_{\mathbf{x},\boldsymbol{\xi}}\left[\|\sqrt{\Delta_t}\mathbf{s_A}(\mathbf{x}_t(\boldsymbol{\xi})) + \boldsymbol{\xi}\|^2\right], \quad \mathcal{E}_{\text{score}} = \frac{1}{d}\mathbb{E}_\mathbf{x}\left[\|\mathbf{s_A}(\mathbf{x}) - \nabla \log P_\mathbf{x}\|^2\right], \tag{8}$$

where the expectations $\mathbb{E}_{\mathbf{x},\boldsymbol{\xi}}$ are computed over $\mathbf{x} \sim P_\mathbf{x}$ and $\boldsymbol{\xi} \sim \mathcal{N}(0, \boldsymbol{I}_d)$. The generalization loss, defined as $\mathcal{L}_{\text{gen}} = \mathcal{L}_{\text{test}} - \mathcal{L}_{\text{train}}$, indicates the degree of overfitting in the model while the distance to the exact score $\mathcal{E}_{\text{score}}$ measures the quality of the generation as it is an upper bound on the Kullback–Leibler divergence between the target and generated distributions [48, 7]. The weights $\mathbf{A}$ are updated via gradient descent

$$\mathbf{A}^{(k+1)} = \mathbf{A}^{(k)} - \eta\nabla_\mathbf{A}\mathcal{L}_{\text{train}}(\mathbf{A}^{(k)}), \tag{9}$$

where $\eta$ is the learning rate. In the high-dimensional limit, as the learning rate $\eta \to 0$, and after rescaling time as $\tau = k\eta/d^2$, the discrete-time dynamics converges to the following continuous-time gradient flow:

$$\dot{\mathbf{A}}(\tau) = -d^2\nabla_\mathbf{A}\mathcal{L}_{\text{train}}(\mathbf{A}(\tau)) = -2\Delta_t\frac{d}{p}\mathbf{AU} - \frac{2d\sqrt{\Delta_t}}{\sqrt{p}}\mathbf{V}^T, \tag{10}$$

with

$$\mathbf{U} = \frac{1}{n}\sum_{\nu=1}^n \mathbb{E}_{\boldsymbol{\xi}}\left[\sigma\left(\frac{\mathbf{Wx}_t^\nu(\boldsymbol{\xi})}{\sqrt{d}}\right)\sigma\left(\frac{\mathbf{Wx}_t^\nu(\boldsymbol{\xi})}{\sqrt{d}}\right)^T\right], \quad \mathbf{V} = \frac{1}{n}\sum_{\nu=1}^n \mathbb{E}_{\boldsymbol{\xi}}\left[\sigma\left(\frac{\mathbf{Wx}_t^\nu(\boldsymbol{\xi})}{\sqrt{d}}\right)\boldsymbol{\xi}^T\right]. \tag{11}$$

**Assumptions.** For our analytical results to hold, we make the following mathematical assumptions which are standard when studying Random Features [37, 15, 20] namely (i) the activation function $\sigma$ admits a Hermite polynomial expansion $\sigma(x) = \sum_{s=0}^\infty \frac{\alpha_s}{s!}He_s(x)$; and (ii) the data distribution $P_\mathbf{x}$ has zero mean, sub-Gaussian tails and a covariance $\boldsymbol{\Sigma} = \mathbb{E}_{P_\mathbf{x}}[\mathbf{x}\mathbf{x}^T]$ with bounded spectrum. We assume that the empirical distribution of eigenvalues of $\boldsymbol{\Sigma}$ converges weakly in the high dimensional limit to a deterministic density $\rho_{\boldsymbol{\Sigma}}(\lambda)$ and that $\text{Tr}(\boldsymbol{\Sigma})/d$ converges to a finite limit (for a more precise mathematical statement see SM Sect. C.3). Moreover, we make additional assumptions that are not essential to the proofs but which simplify the analysis: (iii) the activation function $\sigma$ verifies $\mu_0 = \mathbb{E}_z[\sigma(z)] = 0$ for $z$ standard Gaussian; and (iv) the second layer $\mathbf{A}$ is initialized with zero weights $\mathbf{A}(\tau = 0) = 0$. In numerical applications, unless specified, we use $\sigma(z) = \tanh(z)$ and $P_\mathbf{x} = \mathcal{N}(0, \boldsymbol{I}_d)$.

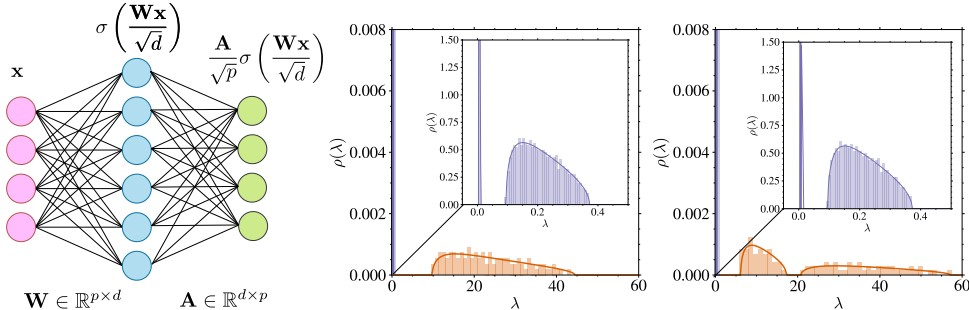

Figure 4: (*Left*) **Illustration of an RFNN.** (*Middle/Right*) **Spectrum of U.** Density $\rho(\lambda)$ from Theorem 3.1 in the overparameterized Regime I described in Theorem 3.2, with $\psi_p = 64$, $\psi_n = 8$, $t = 0.01$, and $\rho_{\boldsymbol{\Sigma}}(\lambda) = \delta(\lambda - 1)$. The bulk of the spectrum (orange) is between $\lambda \approx 10$ and $\lambda \approx 45$. The histogram shows the eigenvalues from a single realization of $\mathbf{U}$ at $d = 100$. Inset: zoom near $\lambda = 0$ (in blue) showing the first bulk $\rho_1$ and the delta peak at $\lambda = s_t^2$. (*Right*) Same as (*Middle*), but with $\rho_{\boldsymbol{\Sigma}}(\lambda) = \frac{1}{2}\delta(\lambda - 0.5) + \frac{1}{2}\delta(\lambda - 1.5)$. The first bulk in blue remains unchanged, as it depends only on $\sigma_{\mathbf{x}}^2 = \mathrm{Tr}(\boldsymbol{\Sigma})/d = 1$ in both cases, while the second bulk varies with $\boldsymbol{\Sigma}$.

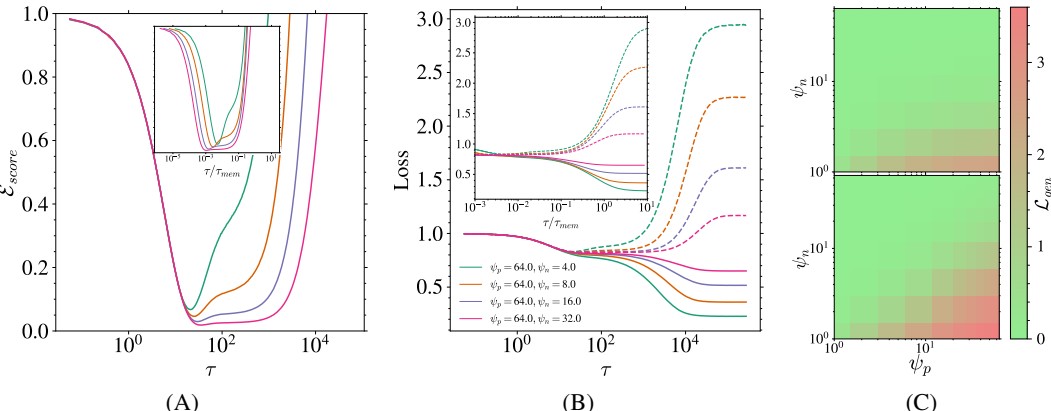

Figure 5: **Evolution of the training and test losses for the RFNN.** (A) Distance to the true score $\mathcal{E}_{\mathrm{score}}$ against training time $\tau$ for $\psi_n = 4, 8, 16, 32, \psi_p = 64, t = 0.1$ and $d = 100$. In the inset, the training time is rescaled by $\tau_{\mathrm{mem}} = \psi_p/\Delta_t\lambda_{\mathrm{min}}$. (B) Training (solid) and test (dashed) losses for various $\psi_n$. The inset shows both losses rescaled by $\tau_{\mathrm{mem}}$. (C) Heatmaps of $\mathcal{L}_{\mathrm{gen}}$ for $\tau = 10^3$ (top) and $\tau = 10^4$ (bottom) as a function of $\psi_n$ and $\psi_p$. All the curves use Pytorch [35] gradient descent. More numerical details can be found in SM Sect. D.

**Emergence of the two timescales during training.** We first show in Fig. 5 that the behavior of training and test losses in the RF model mirrors the one found in realistic cases in Sect. 2, with a separation of timescales $\tau_{\mathrm{gen}}$ and $\tau_{\mathrm{mem}}$ which increases with $n$. Equation (10) is linear in $\mathbf{A}$ and hence it can be solved exactly (see SM). The timescales of the training dynamics are given by the inverse eigenvalues of the $p \times p$ matrix $\Delta_t \mathbf{U}/\psi_p$. Building on the Gaussian Equivalence Principle [GEP, 14, 15, 33] and the theory of linear pencils [6], George et al. (2025) derive a coupled system of equations characterizing the Stieltjes transform of the eigenvalue density $\rho(\lambda)$ of $\mathbf{U}$ for isotropic Gaussian data that lie in a $D$-dimensional subspace with $D \leq d$ and $D = \mathcal{O}(d)$. We offer an alternative derivation presented in SM for general variance using the replica method [34] – a heuristic method from the statistical physics of disordered systems – yielding the more compact formulation for obtaining the spectrum stated in Theorem 3.1. Before stating the theorem, we introduce

$$b_t = \mathbb{E}_{u,v}[v\sigma(e^{-t}\sigma_{\mathbf{x}}u + \sqrt{\Delta_t}v)], \quad a_t = \mathbb{E}_{u,v}[\sigma(e^{-t}\sigma_{\mathbf{x}}u + \sqrt{\Delta_t}v)\frac{u}{e^{-t}\sigma_{\mathbf{x}}}], \tag{12}$$

$$v_t^2 = \mathbb{E}_{u,v,w}[\sigma(e^{-t}\sigma_{\mathbf{x}}u + \sqrt{\Delta_t}v)\sigma(e^{-t}\sigma_{\mathbf{x}}u + \sqrt{\Delta_t}w)] - a_t^2 e^{-2t}\sigma_{\mathbf{x}}^2, \tag{13}$$

$$s_t^2 = \mathbb{E}_u[\sigma(\Gamma_t u)^2] - a_t^2 e^{-2t}\sigma_{\mathbf{x}}^2 - v_t^2 - b_t^2, \tag{14}$$

where $\sigma_{\mathbf{x}}^2 = \frac{\text{Tr}(\mathbf{\Sigma})}{d}$, $\Gamma_t = e^{-2t}\sigma_{\mathbf{x}}^2 + \Delta_t = 1 + e^{-2t}(\sigma_{\mathbf{x}}^2 - 1)$ and the expectation is over the $u, v, w$ random variables which are independent standard Gaussian $\mathcal{N}(0,1)$.

**Theorem 3.1.** *Let* $q(z) = \frac{1}{p}\text{Tr}(\mathbf{U} - z\mathbf{I}_p)^{-1}$, $r(z) = \frac{1}{p}\text{Tr}(\mathbf{\Sigma}^{1/2}\mathbf{W}^T(\mathbf{U} - z\mathbf{I}_p)^{-1}\mathbf{W}\mathbf{\Sigma}^{1/2})$ *and* $s(z) = \frac{1}{p}\text{Tr}(\mathbf{W}^T(\mathbf{U} - z\mathbf{I}_p)^{-1}\mathbf{W})$, *with* $z \in \mathbb{C}$. *Let*

$$\hat{s}(q) = b_t^2\psi_p + \frac{1}{q}, \tag{15}$$

$$\hat{r}(r,q) = \frac{\psi_p a_t^2 e^{-2t}}{1 + \frac{a_t^2 e^{-2t}\psi_p}{\psi_n}r + \frac{\psi_p v_t^2}{\psi_n}q}. \tag{16}$$

*Then* $q(z), r(z)$ *and* $s(z)$ *satisfy the following set of three equations:*

$$s = \int d\rho_{\mathbf{\Sigma}}(\lambda)\frac{1}{\hat{s}(q) + \lambda\hat{r}(r,q)}, \tag{17}$$

$$r = \int d\rho_{\mathbf{\Sigma}}(\lambda)\frac{\lambda}{\hat{s}(q) + \lambda\hat{r}(r,q)}, \tag{18}$$

$$\psi_p(s_t^2 - z) + \frac{\psi_p v_t^2}{1 + \frac{a_t^2 e^{-2t}\psi_p}{\psi_n}r + \frac{\psi_p v_t^2}{\psi_n}q} + \frac{1 - \psi_p}{q} - \frac{s}{q^2} = 0, \tag{19}$$

*The eigenvalue distribution of* $\mathbf{U}$, $\rho(\lambda)$, *can then be obtained using the Sokhotski–Plemelj inversion formula* $\rho(\lambda) = \lim_{\varepsilon \to 0^+} \frac{1}{\pi}\text{Im}\, q(\lambda + i\varepsilon)$.

We now focus on the asymptotic regime $\psi_p, \psi_n \gg 1$, typical for strongly over-parameterized models trained on large data sets. In this limit, the spectrum of $\mathbf{U}$ can be described analytically by the following Theorem 3.2.

**Theorem 3.2** (Informal). *Let* $\rho$ *denote the spectral density of* $\mathbf{U}$.

*Regime I (overparametrized):* $\psi_p > \psi_n \gg 1$.

$$\rho(\lambda) = \left(1 - \frac{1 + \psi_n}{\psi_p}\right)\delta(\lambda - s_t^2) + \frac{\psi_n}{\psi_p}\rho_1(\lambda) + \frac{1}{\psi_p}\rho_2(\lambda).$$

*Regime II (underparametrized):* $\psi_n > \psi_p \gg 1$.

$$\rho(\lambda) = \left(1 - \frac{1}{\psi_p}\right)\rho_1(\lambda) + \frac{1}{\psi_p}\rho_2(\lambda).$$

*where* $\rho_1$ *is an atomless measure with support*

$$\left[s_t^2 + v_t^2\left(1 - \sqrt{\psi_p/\psi_n}\right)^2, \; s_t^2 + v_t^2\left(1 + \sqrt{\psi_p/\psi_n}\right)^2\right],$$

*and* $\rho_2$ *coincides with the asymptotic eigenvalue bulk density of the population covariance* $\tilde{\mathbf{U}} = \mathbb{E}_{\mathbf{X}}[\mathbf{U}]$; $\rho_2$ *is independent of* $\psi_n$ *and its support is on the scale* $\psi_p$. *The eigenvectors associated with* $\delta(\lambda - s_t^2)$ *leave both training and test losses unchanged and are therefore irrelevant. In the limit* $\psi_p \gg \psi_n$, *the supports of* $\rho_1$ *and* $\rho_2$ *are respectively on the scales* $\psi_p/\psi_n$ *and* $\psi_p$, *i.e. they are well separated.*

The proofs of both theorems are shown in SM (Sect. C). We recall that training timescales are directly related to eigenvalues $\lambda$ via the relation $\tau^{-1} = \psi_p/\Delta_t\lambda_{\min}$. Theorem 3.2 therefore demonstrates the emergence of the two training timescales $\tau_{\text{mem}}$ and $\tau_{\text{gen}}$ in the overparametrized regime of the RFNN model. They are respectively associated to the measures $\rho_1$ and $\rho_2$, which are well separated in regime I, for $\psi_p \gg \psi_n \gg 1$, as shown in Fig. 4.

**Generalization**: The timescale $\tau_{\text{gen}}$ on which the first relaxation takes place is associated to the formation of the generalization regime. It is related to the bulk $\rho_2$ and is or order $1/\Delta_t$. This regime only depends on the population covariance $\mathbf{\Sigma}$ of the data and is independent of the specific realization

of the dataset. On this timescale, which is of order one, both the training $\mathcal{L}_{\mathrm{train}}$ and test $\mathcal{L}_{\mathrm{test}}$ losses decrease. The generalization loss $\mathcal{L}_{\mathrm{gen}} = \mathcal{L}_{\mathrm{test}} - \mathcal{L}_{\mathrm{train}}$ is zero, and $\mathcal{E}_{\mathrm{score}}$ tends to a value that we find to scale as $\mathcal{O}(\psi_n^{-\eta})$ with $\eta \simeq 0.59$ numerically (see Fig. 5).

**Memorization:** The timescale $\tau_{\mathrm{mem}}$, on which the second stage of the dynamics takes place, is associated to overfitting and memorization. It is related to the bulk $\rho_1$, and scales as $\psi_p/\Delta_t \lambda_{\min}$, where $\lambda_{\min}$ is the left edge of $\rho_1$. In the overparameterized regime $p \gg n$, $\tau_{\mathrm{mem}}$ becomes large and of order $\psi_n/\Delta_t$, thus implying a scaling of $\tau_{\mathrm{mem}}$ with $n$. On this timescale, the training loss decreases while the test loss increases, converging to their respective asymptotic values as computed in [13]. Fig. 5 indeed shows that all training and test curves separate, correspondingly the generalization loss $\mathcal{L}_{\mathrm{gen}}$ increases, at a time that scales with $\psi_p/\Delta_t \lambda_{\min}$, as shown in the inset.

As $n$ increases, the asymptotic ($\tau \to \infty$) generalization loss $\mathcal{L}_{\mathrm{gen}}$ decreases, indicating a reduced overfitting. For $n > n^*(p) = p$, although some overfitting remains (i.e., $\mathcal{L}_{\mathrm{gen}} > 0$), the value of $\mathcal{L}_{\mathrm{gen}}$ is sensibly reduced, and the model is no longer expressive enough to memorize the training data, as shown in [13]. This regime corresponds to the *Architectural Regularization* phase in Fig. 1. We show in Fig. 5 (panel C) how the generalization loss $\mathcal{L}_{\mathrm{gen}}$ varies in the $(n, p)$ plane depending on the time $\tau$ at which training is stopped. In agreement with the above results, we find that the generalization–memorization transition line depends on $\tau$ and moves upward for larger values of $\tau$, similarly to the numerical results exposed in Fig. 3 and the illustration in Fig. 1.

## 4   Conclusions

We have shown that the training dynamics of neural network-based score functions display a form of implicit regularization that prevents memorization even in highly overparameterized diffusion models. Specifically, we have identified two well-separated timescales in the learning: $\tau_{\mathrm{gen}}$, at which models begins to generate high-quality, novel samples, and $\tau_{\mathrm{mem}}$, beyond which they start to memorize the training data. The gap between these timescales grows with the size of the training set, leading to a broad window where early stopped models generate novel samples of high-quality. We have demonstrated that this phenomenon happens in realistic settings, for controlled synthetic data, and in analytically tractable models. Although our analysis focuses on DMs, the underlying score-learning mechanism we uncover is common to all score-based generative models such as stochastic interpolants [3] or flow matching [29]; we therefore expect our results to generalize to this broader class.

**Limitations and future works.**

- While we derived our results under SGD optimization, most DMs are trained in practice with Adam [25]. In SM Sects. A.3 and D, we show that the two key timescales still arise using Adam, although with much fewer optimization steps. Studying how different optimizers shift these timescales would be valuable for practical usage.

- All experiments in Sect. 2 are conducted with unconditional DMs. We additionally verify in SM Sect. B, using a toy Gaussian mixture dataset and classifier-free guidance [19], that the same scaling of $\tau_{\mathrm{mem}}$ with $n$ holds in the conditional settings. Understanding precisely how the absolute timescales $\tau_{\mathrm{mem}}$ and $\tau_{\mathrm{gen}}$ depend on the conditioning remains an open question.

- Our numerical experiments cover a range of $p$ between 1M and 16M. Exploring a wider range is essential to map the full $(n, p)$ phase diagram sketched in Fig. 1 and understand the precise effect of expressivity on dynamical regularization.

- Finally, our theoretical analysis rely on well-controlled data and score models that reproduce the core effects. Extending these analytical frameworks to richer data distributions (such as Gaussian mixtures or data from the hidden manifold model) and to structured architectures would be valuable to further characterize the implicit dynamical regularization of training score-functions. In particular investigating how heavy-tailed data distribution [2] affect the picture described here could be valuable.

- Although DMs trained on large and diverse datasets likely avoid the memorization regime we study here, some industrial models were shown to exhibit partial memorization [8, 45]. Our results provide practical guidelines (early-stopping, control the network capacity) to train DMs robustly and hence avoid memorization, which can be especially helpful in data-scarce domains (e.g., physical sciences).

## Acknowledgments and Disclosure of Funding

The authors thank Valentin De Bortoli for initial motivating discussions on memorization–generalization transitions. RU thanks Beatrice Achilli, Jérome Garnier-Brun, Carlo Lucibello and Enrico Ventura for insightful discussions. RU is grateful to Bocconi University for its hospitality during his stay, during which part of this work was conducted. This work was performed using HPC resources from GENCI-IDRIS (Grant 2025-A0181016159). GB acknowledges support from the French government under the management of the Agence Nationale PR[AI]RIE-PSAI (ANR-23-IACL-0008). MM acknowledges the support of the PNRR-PE-AI FAIR project funded by the NextGeneration EU program. After completing this work, we became aware that A. Favero, A. Sclocchi, and M. Wyart [12] had also been investigating the memorization–generalization transition from a similar perspective.

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
