# OpenReview forum: "Why Diffusion Models Don’t Memorize:  The Role of Implicit Dynamical Regularization in Training"
_NeurIPS.cc/2025/Conference — NeurIPS 2025 oral_

### Official Review · Reviewer_piNC · 2025-06-16

**Clarity:** 4
**Significance:** 3
**Originality:** 3
**Rating:** 5
**Confidence:** 3

**Summary:**

This work examines the role of early stopping in preventing memorization in diffusion models. They observe that when the number of samples $n$ are increased, the so called window of generalization, the stopping times at which generalization turns into memorization, scales with $n$. They verify this in a toy setting with Celeb A data, and verify this scaling theoretically for random Gaussian data in the random feature model for the score function, deriving results in the high-dimensional limit using the replica method based on previous work.

**Questions:**

Questions

1. Is the phenomenon you observe unique to training diffusion models? Or does it also occur when you simply want to train a model to memorize given input data? It would be great to explicitly comment on this in the manuscript.
2. Fig. 3 right: Generally, this plot is rather confusing, as the visualization suggests that one should to match the two lines to the architectural and dynamical regularization in the theoretical intro Figure 1. But it seems that since you do not have n large / p small enough, the architectural bias curve does not show up. Can you clarify, and edit the figure so that it is not suggesting direct correspondence between theory and experiment lines? (or correct my understanding, if this is the intended case!)
3. Paragraph L.189: It generally seems that experiments on larger p/n would help support your results, you just state your expectation and refer it to future work, yet it seems that it would underline a crucial point about architectural regularization you make in the introduction and that would not be a substantially new direction of study.
4. Optionally, it could be nice to make the memorization scores more tangible by showing images of the closest two samples used in the metric for a random subset of training samples - long with the values of the scores - similar to the FID in the Appendix.
5. Can you clarify whether the assumptions you make in L.239 are mainly technical or due to fundamental limitations? (Similarly for the frozen first layer weights?)
6. Is the code available only on request, or will it be published?

Some small comments
1. L.45 seems like the first time where you introduce $\tau$ - for clarity it could be useful to distinguish it from the time of the diffusion process $t$ and specifiy whether you mean training budget in an online setting, or epochs of the optimization process.
2. L.41 implicit bias — it would help to spell out what this bias is biasing towards (generalizing instead of memorizing solutions?)
3. Personally I find the term “dynamical regularization” slightly misleading, as my first thought was that the regularization would be dynamical, i.e. adaptive. On second thought it had to do with the diffusion time dimensions. Both interpretations were wrong. It could be helpful to say very early in the introduction that early stopping is an example, which both indicates that you mean training dynamics and rules out the diffusion itself.
4. L.152 it could be useful to mention explicitly that the FID is used to measure generalization.
5. Do you have an understanding on how sensitive your results are to your choice of k=1/3 (6)
6. Fig. 3 right: Can you add markers to the phase diagram, so that is clear what the measured datapoints were? Which W is used here? All of them?
7. L.227 From context clear but not easy to parse: Do you rescale time or the loss?
8. In the caption of Fig. 6 could you clarify whether the lines are theoretical or empirical.
9. L.229 it might be worth clarifying that the score is not a function which depends on time, but that the time is a property of the input, i.e. write $s(\mathbf x_t)$.
10. Can you clarify in Figure 5 whether we are in Regime I or II, and what can be identified as $\rho_1$ or $\rho_2$? Or if it is an misunderstanding, why this distribution is not related to the forms in Thm. 3.2?

**Ethical Concerns:**

["NO or VERY MINOR ethics concerns only"]

**Final Justification:**

All questions I had were adressed and with the extra page, I believe that the authors will be able to incorporate them in the given space.

**Limitations:**

They are openly adressed by the authors in a separate section and discussed in weaknesses in this review.

**Paper Formatting Concerns:**

None.

**Quality:**

3

**Strengths And Weaknesses:**

Strengths
1. The exposition of the problem is well-motivated and the manuscript is clearly written.
2. The metrics selected for memorization and generalization in diffusion models are intuitive and easy to compute and interprete.
3. The theoretical results are communicated clearly and extend existing results.

Weakness
1. As mentioned by the authors, larger parameter regimes for the models would be desirable, and they propose this as future work. However, it seems important to the empirical verification of the phase diagram in Fig. 3 to verify these regimes as well. It would be useful to understand if there are deeper reasons why these experiments were not included.
2. It is not entirely clear how much of this analysis is specific to the score function in diffusion models - or if the phenomenon of first generalizing then memorizing is something that is more generally observed in random feature models.
3. In the context of the score function, getting the score correct at different diffusion times $t$ is weighted the same, but errors could accucmulate on sampling new items. Yet, in the experiments with Celeb, the final generated samples are compared. It is unclear to me in how far these measurements are comparable.
4. It is unclear whether the theory that holds for GD is inidicative of adaptive optimizers. For the empirical results on Celeb this could be tested for at least a smaller fraction of hyperparameter values to obtain an idication.

---

> ### Author Rebuttal · Authors · 2025-07-31
>
> We first want to thank you for the very positive report as well as for your different questions that helped us to improve our work. You will find below an answer to the several points raised in your review.
>
> ## Main comments
>
> > **About larger parameters regime for Fig. 3**
>
> We have conducted additional experiments in the meantime. In particular, we added an intermediary point $W=48$ ($p \approx 9$M) and found that for CelebA resized to 32x32 the range of $p\in\left[10^6, 16\times 10^6\right]$ parameters show convergence on the training timescales we focus on ($8\times \tau_\mathrm{gen}$ at maximum) and therefore does not require much larger $p$. We plan to update Fig. 3 and the discussion accordingly in the final version of the manuscript. Exploring the diagram for much larger $p$ on this dataset would demand significantly larger $n$ and much more training time, which goes beyond our computational capacities without, we believe, strengthening our core results (see also the answer below).
>
> > **About Fig. 3 (right) and the architectural regularization**
>
> We agree that although the layouts of Fig. 1 and Fig. 3 are similar, they plot related but distinct diagnostics, which may be confusing. Fig. 1 is a schematic diagram drawn in the $\tau\to\infty$ regime, illustrating the theoretical findings. On the other hand, Fig. 3 fixes $\tau = \ell \tau_\mathrm{gen}$ and empirically identifies the smallest $n(p, \tau)$ at which $f_\mathrm{mem}(\tau)=0$. Building upon your question, we were able to identify, for our range of $p$ and training time (2M steps), convergence of $n(p,\tau)$ as $\tau$ increases, therefore drawing the architectural regularization line $n^\star(p)$ hinted by Fig. 1. We thank you for this suggestion and we will definitely add this line to Fig. 3 to reinforce the parallel with Fig. 1 and validate the observations on realistic trained models and datasets.
>
> > **About generalization to adaptive optimizers**
>
> In the numerical experiments on CelebA we use SGD with fixed momentum. To demonstrate that the observed phenomenon is not specific to SGD, we included in Appendix B.3. numerical experiments on a toy Gaussian Mixture dataset using Adam which exhibit the same behaviour. Since the submission, we also conducted additional experiments on CelebA at fixed $W=64$, and on random features, both trained with Adam, showing that $\tau_\mathrm{mem}$ also scales linearly with $n$ in these cases. We will be pleased to incorporate these new results in the appendix of the camera‐ready version of the manuscript, accompanied with a remark in the conclusion.
>
> > **Is the phenomenon unique to diffusion models?**
>
> Thank you for raising this important question on the generality of our reuslts. We believe that our findings extend to other generative models. For instance, the same analysis can be done transparently for stochastic interpolants [1] and for Flow Matching [2]. Indeed, the learning of the score function (or velocity field) can be rephrased as a regression task (the denoising score-matching loss in the case of diffusion models). In the context of supervised learning, [3] observed a similar timescales for overfitting with an overparametrized two-layer neural network. More specifically, they show that if $n/p$ is kept fixed, then this timescale goes linearly with $p$, which is similar to our result. We will mention this point in the final version of the paper.
>
> > **Making memorization scores more tangible**
>
> We show in Fig. 7 some generated images along with their nearest neighbor in the training set. We will make use of the additional page of the camera-ready version to include a similar (but better-looking) figure in the main text showing a random batch of generated sample making the memorization fractions more tangible (e.g. in Fig. 2).
>
> > **About the assumptions of L239**
>
> We thank you for asking this clarification. It is indeed important to assess the role of these assumptions and simplifications. We will do it in the camera-ready version if the paper is accepted. There are two assumptions made in the analytical part: (i) the activation function $\sigma$ admits an expansion in Hermite polynomials and (ii) $\sigma$ is odd. Both are mainly technical and are not a limitation of the approach. Assumption (i) is equivalent to being in the functional space $$L^2(e^{-x^2/2}dx)=\{f :\int \mathrm{d} x  \lvert f(x)\rvert^2e^{-x^2/2}<+\infty\},$$
> and is verified by all the activation functions used in practice ($\tanh$, ReLU...). Regarding (ii), it is the same as in [4], and greatly simplifies computations. Moreover, the target score function we focus on also has this symmetry, therefore it's natural to consider such families of activation functions. It can be relaxed by requiring only that $\mu_0=\mathbb{E}_{z\sim\mathcal{N}(0,1)}[\sigma(z)]=0$, which simply amounts to a constant shift of the activation function $\tilde{\sigma}=\sigma-\mu_0$.
>
> Focusing on the case where the first layer of weights is frozen is a limitation. It allows us to  study the memorization-generalization phenomenon in a well-defined and solvable setting, the celebrated Random Features model [5]. While simplistic, it has already served as theoretical framework to study several behaviors of deep neural network such as the double descent phenomenon [6,7].
> We want to stress that our approach can be extended to more general settings. In fact, the Neural Tangent Kernel scaling limit of deep neural networks can be mapped to a specific Random Feature model [8,9].
> Therefore, one can extend the approach we developed in this work to study deep neural networks models of the score in the NTK limit - the only difference being that the random matrix problem to solve (Theorem 3.1) is more involved but conceptually identical.
> After the completion of this work we started to work on this project and the associated results will be presented in a forthcoming publication.
>
> We will clarify the nature and importance of each assumption in the final version of the paper.
>
> > **Availability of the code**
>
> Thank you for raising this point. We will make all codes used for training our models and reproducing the main figures of the paper publicly available in a non-anonymized Github repository upon publication.
>
> ## Smaller comments
>
> 1. $\tau$ is the training time, measured in the paper as the number of gradient updates. We will make this clearer in the final version of the paper.
>
> 2. We will make this sentence clearer to avoid any confusion.
>
> 3. We understand that the chosen wording might be misleading at first sight and we are committed to making it clearer at the very beginning of the next version of the paper.
>
> 4. Thank you, we now specify that the evolution of the FID is used to actually measure $\tau_\mathrm{gen}$ in the numerical experiments.
>
> 5. The choice $k=1/3$ comes from previous numerical studies [10, 11] to fit the visual appreciation of memorization on image datasets. We made sure that varying $k$ (e.g. to $1/2$ or $1/4$) does not affect the scaling behaviour of $\tau_\mathrm{mem}$.
>
> 6. We reworked the diagram in Fig. 3 (right) as you suggested to add markers indicating clearly the measured points. We also completed it with a two parameter values with $W=48$ and $W=8$. All available $W$ are used.
>
> 7. In Section 3, we rescale the Loss of Eq. (4) by the dimension $d$ of the data to have a well-defined large dimensional limit. We agree that the formulation of the sentence is ambiguous and we will clarify it in the camera-ready version of the paper.
>
> 8. The lines in Fig. 6 are empirical, obtained using a PyTorch implementation of full-batch gradient descent. Details of the numerical setup can be found in Appendix D. We will make this explicit in the figure caption by indicating that the curves are empirical.
>
> 9. In full generality, the score function $s(x,t)$ is a function of $x\in\mathbb{R}^d$ and the diffusion time $t$. In the analytical part of the article, we fix the diffusion time $t$ and study the learning of the score at this specific time $t$. In other words, we train a new model for the score at each diffusion time $t$. Usually, practitioners only use one model for the whole generative process which depends on the time $t$. However, in our setting, the learned parameters $A$ depend on the diffusion time. Thus the score function depends on time through its parameters $s_A(x)=s_{A(t)}(x)$. This setting was also used in some previous works [4,12] and greatly simplifies the analysis. We thank you for raising it and we will make the dependence of $A$ on time $t$ (and therefore of the score function $s_{A(t)}$) more explicit in the camera-ready version of the article to avoid any confusion.
>
>
> 10. Figure 5 is computed with parameters $\psi_p=64$ and $\psi_n=32$ which corresponds to the Regime I (overparametrized regime) of Thm.3.2. which is the regime of interest of the article. The first bulk in the inset, whose support lies approximately between $\lambda = 0$ and $\lambda = 0.016$, corresponds to the density $\rho_1$ in the theorem. The second bulk, supported between $\lambda = 15$ and $\lambda = 35$, corresponds to $\rho_2$. We thank the reviewer for pointing out the lack of precision in the legend of Figure 5, and we will revise it accordingly in the final version of the paper.
> ----
>
> [1] Albergo, Boffi, Vanden-Eijnden, arxiv: 2303.08797, 2023.
>
> [2] Lipman, Chen, Ben-Hamu, et al., *ICLR*, 2023.
>
> [3] Montanari, Urbani, arxiv:2502.21269, 2025
>
> [4] Georges, Veiga, Macris, arxiv:2502.00336, 2025.
>
> [5] Rahimi, Recht, *Neurips*, 2007.
>
> [6] Mei, Montanari, *Communications on Pure and Applied Mathematics*,2019.
>
> [7] d’Ascoli, Refinetti, Biroli, et al., *ICML*, 2020.
>
> [8] Jacot, Gabriel, Hongler, *NeurIPS*, 2018.
>
> [9] Chizat, Oyallon, Bach. *NeurIPS*, 2019.
>
> [10] Yoon, Choi, Kwon, et al., *ICML*, 2023.
>
> [11] Gu, Du, Pang, et al., *Transactions on Machine Learning Research*, 2025.
>
> [12] Cui, Krzakala, Vanden-Eijnden, et al., *ICLR*, 2025.

---

> > ### Comment · Reviewer_piNC · 2025-08-01
> >
> > Thank you for adressing the points I raised carefully, which will improve the updated version of the work. I have updated my score.

---

> > > ### Author Response · Authors · 2025-08-08
> > >
> > > Thank you for updating your score. We believe that your suggested points will definitely strengthen our manuscript, and we appreciate the care and thoroughness of your review.

---

### Official Review · Reviewer_xUfF · 2025-07-02

**Clarity:** 4
**Significance:** 4
**Originality:** 4
**Rating:** 5
**Confidence:** 2

**Summary:**

This article investigates two important timescales for score-based diffusion models: an early generalization time $\tau_{\mathrm{gen}}$, before which high-quality generation occurs, and a later memorization time $\tau_{\mathrm{mem}}$, after which the generative ability of the diffusion model becomes weakened or limited as the dynamics begin to dominate. The transition regime between these two timescales, related to the so-called dynamical regularization effect, is shown to scale linearly with the training set size $n$, i.e., $\mathcal{O}(n)$. Furthermore, an architectural regularization phase is identified and linked to the expressivity of the underlying neural network.

**Questions:**

Question:
Is there an explicit estimate for the two timescales $\tau_{\mathrm{gen}}$ and $\tau_{\mathrm{mem}}$? How do these timescales depend on the spectrum of $\mathbf{U}$ as described in Theorems 3.1 and 3.2?

**Ethical Concerns:**

["NO or VERY MINOR ethics concerns only"]

**Final Justification:**

The paper is well written and makes strong theoretical contributions. The authors provided clear and satisfactory responses to reviewer concerns.

**Limitations:**

Yes.

**Paper Formatting Concerns:**

None.

**Quality:**

4

**Strengths And Weaknesses:**

Strengths:
The article is clearly written and well organized. It provides a valuable framework for studying one of the most important aspects of diffusion models. The theoretical analysis leverages connections to random matrix theory and spectral analysis, offering a powerful mathematical tool for researchers in the field.

Weakness:
The diffusion model considered in the article is linear, with constant drift and diffusion coefficients (see equation (2)), and the analysis focuses only on one-layer neural network learning (see equation (8)). Although this setup serves well to demonstrate the phenomenon as a toy model, it is limited in capturing nonlinear dynamical effects and in analyzing deep neural networks. Admittedly, extending the analysis to such settings would be a highly challenging task.

---

> ### Author Rebuttal · Authors · 2025-07-31
>
> We first want to thank you for the very positive report concerning the soundness and contributions of our work. You will find below an answer to the points raised in your review.
>
> > **Constant drift and diffusion coefficients in the diffusion model and one-layer neural network for the analytical part.**
>
> We are not completely sure we fully understood the comment, so we will present an extended response and apologize in advance if part of it is not what you referred to.
> The exposition we make of diffusion models in the introduction is indeed kept minimalist, but adding time-dependence to the drift and/or diffusion terms of Eq. 2 in fact amounts to a time reparameterization of the diffusion process. This is therefore not a limitation.  All of our experiments in fact use the standard DDPM formalism with
> $$
> \mathrm{d} \boldsymbol{\mathrm{x}} = -\frac{1}{2} \beta(t) \boldsymbol{\mathrm{x}} \mathrm{d}t + \sqrt{\beta(t)} \mathrm{d}\boldsymbol{B},
> $$
> where $\beta(t)$ is a predefined noise schedule. In Appendix A1, we show that the two formulations are indeed equivalent under a proper reparameterization of the time $t$. Hence, also the theoretical results hold for a non-constant drift and diffusion coefficient.
>
> We do consider a linear score in the theoretical analysis. The motivation is to study the memorization-generalization phenomenon in a well-defined and solvable setting. As you stated, extending the analysis to more general settings would be a highly challenging task. We do hope that our work will trigger new research along these lines. Finally, it is true that we consider a one layer network to model the score, but our approach can be extended to more general settings. In fact,
> the "Neural Tangent Kernel" scaling limit of deep neural networks can be mapped to a specific Random Feature model [1,2].
> Therefore, one can extend the approach we developed in this work to study deep neural networks models of the score in the NTK limit -- the only difference being that the random matrix problem to solve (Theorem 3.1) is more involved but conceptually identical.
> After the completion of this work we started to work on this project. Results will be presented in a forthcoming publication.
>
> We will clarify all these aspects in the final version of the paper to highlight the broad applicability of our results.
>
> > **Is there an explicit estimate for the two timescales $\tau_\mathrm{gen}$ and $\tau_\mathrm{mem}$? How do these timescales depend on the spectrum of $\boldsymbol{U}$ as described in Theorems 3.1 and 3.2?**
>
> As stated in Line 247 of the main text and proved in Proposition C.1 of the SM, the timescales of the training dynamics are given by the inverse eigenvalues of the matrix $\Delta_t \mathbf{U} / \psi_p$, where $\mathbf{U}$ is defined in Eq. (12). In Theorem 3.2, we show that the spectrum of $\mathbf{U}$ exhibits two well-separated bulks, with normalized densities denoted by $\rho_1$ and $\rho_2$. This structure allows us to define two characteristic timescales:
> $$
> \tau_{1} = \frac{\psi_p}{\Delta_t \min[\mathrm{Supp}(\rho_{1})]}\ \ ; \ \ \tau_{2} = \frac{\psi_p}{\Delta_t \min[\mathrm{Supp}(\rho_{2})]}
> $$
>
> Furthermore, Proposition C.2 of the SM shows that on the fast timescale $\tau_2$, both the training and test losses decrease without overfitting, whereas on the longer timescale $\tau_1$, the generalization loss begins to increase. This behavior supports the identification:
> $$
> \tau_{\mathrm{gen}} = \tau_1 \quad \text{and} \quad \tau_{\mathrm{mem}} = \tau_2.
> $$
>
> ----
>
> [1] Jacot, Franck, Clément, Neural tangent kernel: Convergence and generalization in neural networks, NeurIPS, 2018.
>
> [2] Chizat, Oyallon, Bach, On lazy training in differentiable programming, NeurIPS, 2019.

---

> > ### Comment · Reviewer_xUfF · 2025-08-06
> >
> > I would like to thank the authors for addressing my questions. I will maintain my current score (Accept).

---

> ### Author Response · Authors · 2025-08-08
>
> Thank you for your answer and for maintaing your recommendation about the acceptance of our manuscript. We are grateful for your feedback and pleased that our rebuttal addressed your questions.

---

### Official Review · Reviewer_hfaQ · 2025-07-03

**Clarity:** 2
**Significance:** 3
**Originality:** 3
**Rating:** 5
**Confidence:** 3

**Summary:**

This paper investigates the generalization ability of diffusion model, and identifies two crucial time points: an early time when models begin to generate high-quality samples (generation time), and a later time when memorization emerges (memorization time). The generation time is a constant and the memorization time scales with the data size. The findings reveal a form of implicit dynamical regularization, and backed up with both theoretical and empirical investigations.

**Questions:**

It seems the results are restrictive to SGD. Is there a similar phenomena for other gradient based training algorithms with momentum terms?

Are the results applicable to conditional diffusion models?

Is the result applicable on datasets with long-tail distributions? How does the identified implicit dynamic regularization effect affect the learning from samples in long-tail?

**Ethical Concerns:**

["NO or VERY MINOR ethics concerns only"]

**Final Justification:**

Most of my concerns are addressed, I therefore increase my score and recommend for acceptance.

**Limitations:**

yes

**Quality:**

3

**Strengths And Weaknesses:**

The paper is well-written in genaral and easy to follow. This work contributes to the theoretical understanding of generalization performance of diffusion models. I didn't go through the proofs, but the result is sound. The theoreical investigation is based on a random feature simplification. Could the authors provide more elaboration on how that's closely related to the more practical models or what's used in experiments?

More discussions on how such findings help design better training strategies for diffusion models would enhance the significance of this work.

---

> ### Author Rebuttal · Authors · 2025-07-31
>
> We thank you for your remarks and questions that, we believe, will help improve the quality of the paper. Please find below the answers to your interrogations.
>
> > **Discussions about the practical impact of our work**
>
> Several recent work show that even industrial models [1, 2] trained on millions of data exhibit partial of complete memorization of some training samples. While our work is proposing a theoretical understanding of the memorization phenomenon, we believe it also provides interesting guidelines for practitioners on some ways of fixing this issue: either by reducing the total number of parameters in the network as long as it does not harm generation's quality, or by stopping the training earlier. Our work also shows that some intuitive and simple metrics can be used to monitor the amount of memorization during training (memorized fraction or train vs test loss at fixed diffusion times). We will be happy to discuss more how our results could better guide practice in the final version of the paper.
>
> > **Generalization to other optimization algorithms with momentum terms**
>
> In all the experiments on CelebA reported in Sect. II, we in fact use stochastic gradient descent (SGD) with fixed momentum $\beta=0.95$. To demonstrate that the observed phenomenon is not specific to SGD, we included in Appendix B.3. numerical experiments on a toy Gaussian Mixture dataset using Adam which exhibit the same behaviour. Moreover, since the submission, we also conducted additional experiments on CelebA at fixed $W=64$, and on random features, both trained with Adam, showing that $\tau_\mathrm{mem}$ also scales linearly with $n$ in these cases. We will be pleased to incorporate these new results in the appendix of the camera‐ready version of the manuscript, accompanied with a remark in the conclusion.
>
> > **Are the results applicable to conditional diffusion models?**
>
> Thank you for raising this interesting point. We know memorization is still observed in models trained conditionally, as for instance shown in [2, 3, 4]. Importantly, our results do *not* rely on the model being unconditional: any conditioning variable (class, text embeddings or other information) usually enters the model as extra input at the training level. In consequence, our observations are expected to hold when training conditional scores, in particular for each conditioning, we do expect $\tau_\mathrm{mem}$ to increase with $n$. It could happen that $\tau_\mathrm{gen}$ and $\tau_\mathrm{mem}$ depend on the class. In this case, before entering a full memorization (generalization) phase, one could be in a regime of $n$ and $p$ in which some classes are in the generalization (memorization) phase and others do not.
>
> To validate our results on conditional diffusion models, we trained DDPMs with classifier free-guidance [5] on a mixture of two Gaussian (same data and model as in the appendix) and measured the associated memorization fraction during training for several dataset sizes $n$. We show that, even under guidance, $\tau_\mathrm{mem}$ scales linearly with $n$. We will add this result to the appendix and explicitly mention it in the main text of the camera-ready version of the paper as we think it is a valuable addition showing the generality of our results.
>
> > **Is the result applicable on datasets with long-tail distributions?**
>
> Thank you for this question. Since the submission of this work, we have extended our analysis to compute the spectrum of $\mathbf{U}$ for Gaussian data with arbitrary covariance matrix $\Sigma$, including the cases where the density of eigenvalues of $\Sigma$ has power-law tails. We will add this computation in the final version of the paper. The case corresponding to non-Gaussian data, whose distribution has power law tails is more challenging. We have done preliminary numerical experiments on the RF model applied to datasets with long-tail distribution as in [6]. Our findings confirm that the increase of $\tau_\mathrm{mem}$ with $n$  is present also in this case.
>
> A thorough understanding of how heavy-tailed data distributions affect the implicit regularization dynamics would require a more detailed analysis, which goes beyond the current scope of the paper. However, we agree with you that this is an interesting direction that we would like to pursue in a future work.
>
> ----
>
> [1] Carlini, Hayes, Nasir, et al., Extracting Training Data from Diffusion Models, *USENIX*, 2023.
>
> [2] Somepalli, Singla, Goldblum, et al., Understanding and Mitigating Copying in Diffusion Models, *NeurIPS*, 2023.
>
> [3] Wen, Liu, Chen, et al., Detecting, explaining and mitigating memorization in Diffusion Models, *ICLR*, 2024.
>
> [4] Chen, Yu, Xu, Towards Memorization-Free Diffusion Models, arxiv:2404.00922.
>
> [5] Ho, Salimans, Classifier-Free Diffusion Guidance, *NeurIPS*, 2021.
>
> [6] Adomaityte, Defilippis, Loureiro, et al., High-dimensional robust regression under heavy-tailed data: asymptotics and universality, *Journal of Statistical Mechanics: Theory and Experiment*, 2024.

---

> > ### Comment · Reviewer_hfaQ · 2025-08-06
> >
> > Thanks for the detailed reply. Most of my concerns are addressed, and I increased my score accordingly (from weak accept to accept).

---

> > > ### Author Response · Authors · 2025-08-08
> > >
> > > Thank you for informing us about the score increase, we truly appreciate it. We are confident that the modifications you suggested will further enhance the impact of our work.

---

### Official Review · Reviewer_f3Fk · 2025-07-03

**Clarity:** 4
**Significance:** 4
**Originality:** 4
**Rating:** 5
**Confidence:** 5

**Summary:**

This work investigates how diffusion models transition from generalization to memorization. The authors identify two training timescales: an early time $\tau_{gen}$ when models begin generating new samples, and a later time $\tau_{mem}$, when the model starts to memorize training data. The authors claim that the desired generalization behavior of diffusion models occur when the training time is between $\tau_{gen}$ and $\tau_{mem}$. The authors provide experimental evidence and theoretical analysis in the setting of a random feature network.

**Questions:**

Is there a way the findings and the insights of this paper may guide us to change something with the current diffusion model practice?

**Ethical Concerns:**

["NO or VERY MINOR ethics concerns only"]

**Final Justification:**

The main finding of this work is novel and interesting. With the solid experimental and theoretical evidence, the paper provides clear value.

**Limitations:**

yes

**Quality:**

4

**Strengths And Weaknesses:**

Strength
 - The perspective that the model achieves generalization and then eventually memorization as the training progresses is quite compelling. To the best of my knowledge, this is a novel perspective and the experimental evidence seems sound.
 - The characterization that $\tau_{gen}$ is roughly independent of $N$ while $\tau_{mem}$ is proportional to $N$ is a non-obvious and interesting observation.
 - The theoretical analysis with random feature network is convincing.

Weakness
 - Although I really appreciate the conceptual insight that this work offers, I believe that all diffusion models in practice never reach $\tau_{mem}$ or are within the architectural regularization regime. Therefore, the present theory does not inform us as to what to do differently.

---

> ### Author Rebuttal · Authors · 2025-07-30
>
> We thank you for the very supportive report.
>
> > **I believe that all diffusion models in practice never reach  or are within the architectural regularization regime. Is there a way the findings and the insights of this paper may guide us to change something with the current diffusion model practice?**
>
> We agree that models trained on massive and richly diverse datasets operate above the architectural regularization threshold, and are therefore not concerned by the presented theoretical findings. However, even some models industrially trained on massive datasets like LAION were found to exhibit complete or partial memorization [1, 2]. There are also domains where training data are scarce - for instance many physical science applications like cosmology or climate science - hence often falling in a low-data and high-capacity regime.
> In all these cases, our work offers some concrete guidelines and possible fixes (early-stopping and/or controlling the network capacity) that could help to robustly train the diffusion model to avoid memorization without needing more domain-specific inductive biases. We would be happy to discuss these aspects in the conclusion of the camera-ready version of the paper.
>
> ---
>
> [1] Carlini, Hayes, Nasir, et al., Extracting Training Data from Diffusion Models, USENIX, 2023.
>
> [2] Somepalli, Singla, Goldblum, et al., Understanding and Mitigating Copying in Diffusion Models, NeurIPS, 2023.

---

> > ### Comment · Reviewer_f3Fk · 2025-08-08
> >
> > Thank you for the response.
> >
> > In the rebuttal, the authors claim "concrete guidelines and possible fixes (early-stopping and/or controlling the network capacity) that could help to robustly train the diffusion model to avoid memorization", but I believe this qualitative insight was known prior to this paper.
> >
> > In any case, I am happy with the paper, so I maintain my score of acceptance.

---

### Decision · Program_Chairs · 2025-09-17

**Decision:**

Accept (oral)

**Comment:**

This paper makes a significant and cohesive contribution to the theoretical and empirical understanding of generalization and memorization in diffusion models, which are well acknowledged by the reviewers (in both pre-rebuttal and post-rebuttal phases). Empirically, through rigorous and well-designed experiments on the CelebA dataset, the authors clearly demonstrate the emergence of two distinct training timescales: an early timescale $\tau_{\text{gen}}$ for achieving high-quality generation, which remains constant with dataset size $n$, and a later memorization timescale $\tau_{\text{mem}}$, which scales linearly with $n$. This creates an expanding window for effective generalization, a finding that is robustly validated across varying model capacities and supported by comprehensive metrics including FID, memorization fraction, and train/test loss dynamics. The empirical findings benefit further study of the memorization effect of the diffusion model.

Theoretically, the authors provide a complementary and equally rigorous analysis using a tractable random features model, deriving the spectral properties of the feature correlation matrix in the high-dimensional limit. By leveraging tools from random matrix theory, they formally link the eigenvalue distribution to the separation of timescales, confirming the linear scaling of $\tau_{\text{mem}}$ with $n$ and elucidating the role of model and data complexity ratios ($\psi_p$, $\psi_n$). The synergy between clear empirical demonstrations and a solid theoretical foundation offers profound insights into the mechanisms of implicit dynamical regularization.

Overall, the paper is exceptionally well-written and easy to follow, the rebuttal thoroughly addressed all reviewer concerns, and the findings are of fundamental importance to the machine learning community. Thus, I would recommend an oral presentation.